# Impedimetric Determination of Kanamycin in Milk with Aptasensor Based on Carbon Black-Oligolactide Composite

**DOI:** 10.3390/s20174738

**Published:** 2020-08-21

**Authors:** Tatiana Kulikova, Vladimir Gorbatchuk, Ivan Stoikov, Alexey Rogov, Gennady Evtugyn, Tibor Hianik

**Affiliations:** 1A.M. Butlerov’ Chemistry Institute of Kazan Federal University, 420008 Kazan, Russia; TaNKulikova@kpfu.ru (T.K.); leongard87@mail.ru (V.G.); Ivan.Stoikov@mail.ru (I.S.); 2Interdisciplinary Center of Analytical Microscopy of Kazan Federal University, 420008 Kazan, Russia; alexeyrogov111@gmail.com; 3Analytical Chemistry Department, Chemical Technology Institute, Ural Federal University, 620002 Ekaterinburg, Russia; 4Department of Nuclear Physics and Biophysics, Comenius University, 842 48 Bratislava, Slovakia

**Keywords:** electrochemical biosensor, electrochemical impedance spectroscopy, kanamycin, thiacalix[4]arene, milk contamination

## Abstract

The determination of antibiotics in food is important due to their negative effect on human health related to antimicrobial resistance problem, renal toxicity, and allergic effects. We propose an impedimetric aptasensor for the determination of kanamycin A (KANA), which was assembled on the glassy carbon electrode by the deposition of carbon black in a chitosan matrix followed by carbodiimide binding of aminated aptamer mixed with oligolactide derivative of thiacalix[4]arene in a *cone* configuration. The assembling was monitored by cyclic voltammetry, electrochemical impedance spectroscopy, and scanning electron microscopy. In the presence of the KANA, the charge transfer resistance of the inner interface surprisingly decreased with the analyte concentration within 0.7 and 50 nM (limit of detection 0.3 nM). This was attributed to the partial shielding of the negative charge of the aptamer and of its support, a highly porous 3D structure of the surface layer caused by a macrocyclic core of the carrier. The use of electrostatic assembling in the presence of cationic polyelectrolyte decreased tenfold the detectable concentration of KANA. The aptasensor was successfully tested in the determination of KANA in spiked milk and yogurt with recoveries within 95% and 115%.

## 1. Introduction

Antibiotics of bacterial origin are widely applied in human and veterinary medicine [1]. Although their use has revolutionized the treatment of many diseases, antimicrobial resistance has been found since the very beginning of antibiotics therapy as a result of drug overuse or misuse [2,3]. The loss of antibiotic efficiency results in more than 700,000 annual deaths worldwide [4] and up to 10 million deaths worldwide by 2050 is assumed [5]. Aside from human health, antibiotic contamination of meat and milk can cause economic losses due to undesired influence on the fermentation process and possible accumulation in the consumers’ body [6]. Starting from mastitis treatment in dairy cows, antibiotics have become an indispensable part of the modern livestock sector. This is especially true for kanamycin A (KANA), an aminoglycoside antibiotic (see Figure 1 for structural formula), where the maximal residue level (MRL) in milk is limited in the European Union by 150 μg/kg [7]. High KANA levels in plasma cause hearing and balance problems, renal toxicity, and even neuromuscular blockade for people with an allergy to aminoglycosides.

Growing levels of milk and meat contamination with various antibiotics calls for the development of simple and reliable analytical devices intended for on-site assessment of antibiotic residues in food and water. Traditional methods for antibiotics analysis involve high performance liquid chromatography (HPLC) [8,9], gas chromatography [10], capillary electrophoresis [11,12], surface enhanced Raman spectroscopy [13], and ion mobility spectroscopy [14]. Being sensitive and selective, such methods require expensive and sophisticated equipment, well-trained staff, and time- and labor-consuming sample treatment.

Sensor technologies offer alternatives to the preliminary testing of samples and the detection of high levels of contamination. Such technologies are mainly based on biochemical recognition principles and utilize antibodies and aptamers specific to the analyte. Several biosensors have been developed during the past decade for kanamycin determination. Thus, an immunosensor for KANA determination based on glassy carbon electrode (GCE) covered with graphene, porous Au, and Prussian blue in a chitosan matrix has been described [15]. The reaction of the analyte with specific antibodies immobilized by electrostatic adsorption was monitored by changes in the redox properties of the modifying layer. A similar approach was utilized for the determination of the KANA residues in foods for animals [16]. Here, a redox active layer consisted of graphene flakes, thionine in nafion film decorated with Pt nanoparticles. Specific antibodies were physically adsorbed on the layer from the aqueous solution. The thionine signal was also used in the immunosensor including Ag@Fe_3_O_4_ nanoparticles, providing higher amounts of antibodies immobilized [17] and the immunosensor was applied for pork meat testing.

Aptamers are synthetic DNA or RNA oligonucleotides obtained from the random nucleotide library by a combinatorial chemistry approach and then selected against an analyte by affinity chromatography [18] and are considered as an alternative to traditional antibodies. In comparison to antibodies, aptamers offer the following advantages: rather simple manufacture, easy modification, higher chemical and thermal stability in storage, and operation in the assembly of aptasensors [19]. Regarding KANA, several aptasensors have been described [20,21,22,23,24,25,26,27,28,29,30,31,32]. In them, specific aptamers have mostly been immobilized via terminal amino- or thiol-groups on appropriate carriers (carbon nanotubes, metal nanoparticles, Au electrode surface). The reaction with KANA results in the aptamer folding and in increased resistance of the redox indicator transfer. The response is quantified using voltammetry or electrochemical impedance spectroscopy (EIS). Thus, a label-free EIS aptasensor for the KANA detection was reported by Sharma et al. [21]. In this work, KANA specific aptamers were immobilized on the surface of screen-printed carbon electrodes modified by 4-carboxyphenyl radicals. The sensitivity can be additionally enhanced by labeling the aptamer with horseradish peroxidase [22], ferrocene [23,25], and Methylene blue [26,27]. Combination of the aptamer-analyte binding with electrochemiluminescent luminol oxidation [24] and the strand displacement protocol [25,26,27,28] were also described. In them, the aptamer was hybridized with auxiliary DNA sequence, which was released after the KANA binding. As a result, the terminal label moves closer to the electrode and its signal increases with the analyte concentration. The architecture and measurement protocol of such aptasensors vary depending on the structure of the aptamer and auxiliary DNA strand (hairpin DNA, stem-loop probe, or partially hybridized DNA-aptamer hybrids).

In most of these aptasensor platforms based on displacement protocols and other biochemical amplification approaches, additional measurement stages and expensive reagents are required. Together with the necessity of the sophisticated modification of aptamers and auxiliary DNA strands, this complicates both the aptamer assembling and measurement protocol. Furthermore, many manipulations with biochemical enhancement schemes require qualified staff and special conditions of fulfillment. Most of them should be performed only in a well-equipped laboratory. This limits the potential area of application of such biosensors. In the case of fluorescent [30,31] and photolectrochemical detection [32], the measurement protocol involves several steps with the intermediate addition of auxiliary reagents and has some limitations in the analysis of inhomogeneous and turbid samples.

Recently, we have proposed an universal platform for the electrochemical sensing of specific interactions with DNA (DNA aptamer) that take place on the electrode modified with oligolactides bearing the thiacalix[4]arene core [33]. High porosity of the polymer layer caused by macrocyclic fragments of its structure and terminal carboxylate groups made it possible to form ordered polyelectrolyte complexes with DNA and aptamer and obtain well reproducible and sensitive response toward anthracycline drugs and aflatoxin M1.

In this work, we suggested a similar platform to assemble an impedimetric aptasensor to KANA. To improve the performance of the aptasensor, carbon black (CB) was additionally implemented in the surface layer to establish electric wiring of the aptamer, diminish resistance of the underlying layer, and make more reliable immobilization of the aptamers. CB has found increasing attention in electrochemical biosensors in the past decade [34,35] due to low cost, easy deposition, high surface-to-volume ratio, and compatibility with many biomolecules like enzymes [36,37], antibodies [38], and DNA [39]. The CB application in electrochemical sensors and biosensor design has offered many advantages like low working potential, higher loading of the bioreceptor, bigger signal, and rather simple assembling protocol [40]. For the first time, a composite consisted of the CB, chitosan, and thiacalix[4]arene bearing oligolactide fragments was used as support for the aptamer immobilization. The formation of highly porous layer made it possible to increase the quantities of the aptamer attained in the layer and preserve its high permeability for small molecules. The interaction with the analytes resulted in a decrease in the charge transfer resistance on the inner interface of the layer. To the best of our knowledge, only increased resistance was previously observed in the label free aptasensors. The peculiarities of the surface layer structure and of the signal transduction made the aptasensor less sensitive toward matrix compounds of the samples tested.

## 2. Materials and Methods

### 2.1. Reagents

Carbon black (CB) was provided by Imerys (Paris, France). Thiacalix[4]arene in the *cone* configuration bearing oligolactide fragments and denoted as OLA-*cone* was synthesized at the Organic Chemistry Department of Kazan Federal University. Briefly, 5,11,17,23-tetra-*tert*-butyl-25,26,27,28-tetrakis[(hydroxycarbonyl)methoxy]-thiacalix[4]arene in the *cone* configuration was first synthesized as described in [41]. Oligolactic acid was obtained by heating lactic acid in a rotary evaporator, 270 rpm, at 180 °C and 20–40 mbar. Then, the reactants were mixed and heated in an argon atmosphere for four hours in the presence of 0.01% Sn dioctate as the catalyst. This provided an extension of the oligolactide fragments to 13–17 monomer units from the 5–6 fragments obtained in the reaction of the carboxylated thiacalix[4]arene and lactic acid. The structure of the product was earlier confirmed by the NMR ^1^H spectra and MALDI TOF spectroscopy [42]. The reaction scheme is presented in the Appendix A.

The aminated 22-mer aptamer against KANA 3′-NH_2_-TGG-GGG-TTG-AGG-CTA-AGC-CGA-C-5′ described by Zhu et al. [43] was synthesized by Eurogentec (Liège, Belgium). Prior to use, the aptamer solution was activated by thermostating at 90 °C for 3 min, followed by conditioning at 0 °C for 5 min for proper folding of its structure. Kanamycin A (KANA) sulfate from *Streptomyces kanamyceticus* (M.w. 582.58 g/mol), [Ru(NH_3_)_6_]Cl_3_, *N*-(3-dimethylaminopropyl)-*N*′-ethylcarbodiimide (EDC), *N*-hydroxysuccinimide (NHS), poly(diallyldimethylammonium chloride) (20% wt., PDDA), 2-amino-2-(hydroxymethyl)- 1,3-propanediol (Tris), 2-(*N*-morpholino) ethanesulfonic acid (MES), and chitosan were purchased from Sigma-Aldrich (Steinheim, Germany). All other reagents were of analytical grade and used without additional purification. Millipore^®^ water was used for the preparation of working solutions. Milk and yogurt were purchased from the local market.

Electrochemical measurements were performed at pH = 7.0 in 25 mM phosphate buffer (PB) consisting of 24.7 mM Na_2_HPO_4_ and 0.3 mM NaH_2_PO_4_. The pH was adjusted by 1.0 M NaOH and 1.0 M HCl. As a supporting electrolyte, 0.1 M KCl was used. Voltammetric measurements were performed in the presence of 0.1 M K_3_[Fe(CN)_6_] and impedimetric measurements in the presence of 0.01 M K_3_[Fe(CN)_6_] and 0.01 M K_4_[Fe(CN)_6_].

### 2.2. Apparatus

Voltammetric experiments were carried out in direct current (DC) mode with the CHI Electrochemical Workstation 660E (CH Instruments, Inc., Austin, TX, USA) at ambient temperature in a 5 mL working cell equipped with the aptasensor assembled on the GCE (1.7 mm in diameter) as the working electrode with Ag/AgCl/3.0 M KCl as a reference electrode (CHI129, CH Instruments) and Pt wire as a counter electrode.

The EIS spectra were recorded with the amplitude of the potential of 5 mV and the frequency varied from 0.04 Hz to 100 kHz with 30 points. The calculation of the EIS parameters was performed with the NOVA software (Metrohm Autolab b.v., Utrecht, the Netherlands) by fitting data obtained with the *R(RC)(RC)* equivalent circuit.

The scanning electron microscopy (SEM) microimages were obtained with the high-resolution field emission scanning electron microscope Merlin™ (Carl Zeiss, Jena, Germany).

### 2.3. Aptasensor Fabrication

The GCE was first polished with 0.05 μm alumina powder, washed, and then electrochemically cleaned in 0.2 M H_2_SO_4_ by multiple cycling of the potential. Ten to fifteen cycles were recorded in DC mode until stabilization of the voltammograms. Then, 1.35 mg of the CB was suspended by sonication in 1 mL of 0.275% chitosan in 0.05 M HCl. After that, 2 μL of the CB suspension and 1 μL of 1.0 M NaOH were drop-casted on the GCE surface and left to dry at 50 °C for 20 min. Excessive NaOH was washed out with deionized water and the electrode was activated by 60 min incubation in 2 µL of the mixture of 100 mM EDC and 25 mM NHS in 0.05 M MES, pH = 5.5. After that, the mixture was prepared from the OLA-*cone* solution (0.1 mg in 1 mL of 20 mM PB, pH = 7.4) and 1 μM aptamer in 10 mM Tris-HNO_3_ containing 1.0 mM ethylenediaminetetraacetic acid (EDTA). Below, the mixtures are denoted in accordance with the mixing ratio (*v*/*v*): 2:1, 1:1, and 1:2. Two μL of the mixture was placed on the GCE surface and incubated for 20 min. After that, the aptasensor was rinsed with deionized water, dried at ambient temperature, and used for the KANA determination. If not used, the aptasensor was stored in dry conditions at 4 °C. The developed aptasensors retained their main operation characteristics during two months of storage.

The outline scheme of the aptasensor assembly is presented in Figure 2.

Alternatively, the aptasensor based on PDDA was assembled by consecutive deposition of 2 μL of the OLA-*cone* solution (11.4 mg in 20 mL of 20 mM PB, pH = 7.4), 2 μL of 0.1 mg/mL PDDA, and 20 μL of 1 μM aptamer in 10 mM Tris-HNO_3_ containing 1.0 mM EDTA. Each layer was heated in an oven to 50 °C for 10 min. Excess reagent was removed by washing with deionized water.

### 2.4. Kanamycin A Measurements and Real Sample Assay

The aptasensor was fixed upside down and 2 µL of the drug solution was placed on its surface. The electrode was capped with an Eppendorf plastic tube to prevent drying and left for 20 min. Then, it was washed with deionized water and the EIS measurements were performed. The charge transfer resistance was calculated by fitting the Nyquist diagram.

Spiked samples of the milk or yogurt were first thermostated at 40 °C for 30 min, then diluted with methanol in a 1:3 *v/v* ratio and centrifuged at 5000 rpm for 5 min. The supernatant was separated from the sediment and 10-fold diluted with the PB [21]. The aptasensor was incubated for 20 min and its analytical performance was investigated.

## 3. Results and Discussion

### 3.1. Assembly of the Aptasensor

#### 3.1.1. Voltammetric Measurements

The deposition of CB is mostly performed from its suspension in chitosan or dimethylformamide as binders required for the formation of a more stable film [35]. In our case, we preferred to use chitosan. Its positive charge related to the protonated amino groups could compensate for the negative charge of the OLA-*cone*. The adsorption of the surface layer components was monitored using cyclic voltammetry in the presence of [Fe(CN)_6_]^3−/4−^ as redox indicator. The voltammogram contained a symmetrical pair of redox peaks corresponding to the reversible electron exchange (see Appendix A). The peak currents were proportional to the square root from the potential scan rate in accordance with Equation (1) for anodic peaks (see also Appendix A for the 1:2 mixture as an example). This indicates diffusional control of the electron transfer. As can be seen from the equations, the conditions for the transfer of the redox indicator were about the same for the 2:1 and 1:1 mixture though the 1:2 mixture showed increased diffusional resistance of the transfer.
(1)$$2:1\;\mathrm{mixture}:\;I_{p}\;(\mu A)\;=\;\left(10\pm 1\right)+\left(205\pm 5\right)\nu^{1/2}\;(V/s),\;R^{2}\;=\;0.9981,\;n\;=\;7\;1:1\;\mathrm{mixture}:I_{p}\;(\mu A)\;=\;\left(11\pm 2\right)+\left(195\pm 5\right)\nu^{1/2}\;(V/s),\;R^{2}\;=\;0.9956,\;n\;=\;7\;1:2\;\mathrm{mixture}:I_{p}\;(\mu A)\;=\;\left(14\pm 2\right)+\left(109\pm 2\right)\nu^{1/2}\;(V/s),\;R^{2}\;=\;0.9956,\;n\;=\;7$$

For the low content of the aptamer in the mixture, the shape of the peaks had not been significantly changed after deposition of the CB and OLA-*cone*. Increased quantities of the aptamers changed the peak shape due to a higher content of the nonconductive components in the surface layer (compare Appendix A).

Incubation in the KANA solution did not significantly alter the peak potentials but increased appropriate currents. This might be due to the folding of the aptamer in the complex with KANA and partial shielding of the negative charge of the phosphate residues of the aptamer backbone. As a result, the electrostatic repulsion of the redox indicator decreased.

Deposition of the CB suspension increased the effective surface of the GCE due to implementation of conductive particles in the surface layer. The roughness coefficient was estimated from the Randles-Sevcik equation (Equation (2)) [44], where *I_p_* is the cathodic peak current, *n* is the number of electrons transferred, *D* is the ferricyanide diffusion coefficient (*D* = 7.6 × 10^−6^ cm^2^/s [45]), ν is the scan rate, and *c* is the concentration of ferricyanide ions.
(2)$$I_{p}=2.66\cdot 10^{5}n^{2/3}A_{W}D^{1/2}\nu^{1/2}c$$

We have found that the surface area increased 1.3 times in comparison with bare GCE (real surface 2.54 mm^2^). The amount of CB casted was chosen from the criterion of the full electrode coverage and mechanical stability of the coating. Higher quantities resulted in partial leaching of the CB particles during the aptasensor operation whereas lower quantities left bare electrode areas and worsened the metrological characteristics of the response. Other parameters (i.e., pH of the buffer solution or electrolyte content) did not affect the deposition of CB on the GCE (see Appendix A).

The surface concentration of the aptamer in the sensing layer was estimated using the [Ru(NH_3_)_6_]^3+^ redox indicator [46]. Contrary to [Fe(CN)_6_]^3−/4−^, the Ru complex reacts quantitatively with the DNA nucleobases and remains electrochemically active after accumulation in the aptamer layer. After that, the aptasensor was transferred into the PB with no indicator and the charge transferred in the redox conversion of [Ru(NH_3_)_6_]^3+^ was measured. The aptamer surface density $\Gamma_{Apt}$ was determined from the charge Q measured by coulometry after transfer of the aptasensor with accumulated indicator in the PB with no [Ru(NH_3_)_6_]^3+^ (Equations (3) and (4)).
(3)$$\Gamma_{Ru}=Q/nFA_{W}$$
(4)$$\Gamma_{Apt}=\Gamma_{Ru}N_{A}Z/m$$
where *m* is the number of nucleotides in the aptamer chain (*m* = 30); *Z* is the charge of the redox indicator (*Z* = 3); and *N_A_* is Avogadro’s number. The Γ*_Apt_* was found to be (8 ± 1) × 10^13^ molecules per cm^2^. This is about tenfold higher than that obtained on a flat Au electrode with the monolayer of the aptamers [25]. This confirms the suggestion that the 3D architecture of the surface layer makes it possible to accumulate higher amounts of the aptamers stabilized in the pores of the support. The calculation of the surface density was performed for the casting of 2 μL of 1:2 mixture of the OLA-*cone* and 1.0 μM aptamer solution containing 1.4 × 10^−12^ mole of the aptamers. Thus, practically all aptamer molecules were accumulated in the layer. Changes in the quantities of the aptamers by utilizing other mixtures (2:1, 1:1) gave correspondingly changing values of the $\Gamma_{Apt}$ ((5.0 ± 0.5) × 10^13^, and (6.2 ± 0.5) × 10^13^, respectively). Thus, the experiments with the [Ru(NH_3_)_6_]^3+^ redox indicator confirmed the attachment of the aptamer to the underlying layer and the voltammetric experiments with [Fe(CN)_6_]^3−/4−^ revealed high permeability of the surface layer for small ions. The latter is especially important for the following binding of rather small hydrophilic KANA molecules.

#### 3.1.2. SEM Measurements

SEM images (Figure 3) confirmed the formation of a dense film of chitosan with implemented CB particles. Contrary to the deposition of the CB from DMF suspension, the use of chitosan as a film forming material smoothened the shape of the CB particles.

The addition of the OLA-*cone*/aptamer mixture resulted in the formation of highly porous 3D structure with rather big and tangled channels. This coincided well with similar investigations of the oligolactide layer after cathodic deposition of elemental silver: metal particles formed dendritic palm tree like structures with a high number of branching and a high variation of dimensions [47]. The formation of silver dendrites was affected by the asymmetrical structure of the thiacalix[4]arene in a *cone* macrocyclic core with all the substituents positioned from one side of the macrocycle core. Such a structure of the aptamer layer explains the low level of diffusional limitations observed on cyclic voltammograms of the redox indicator ([Fe(CN)_6_]^3-/4−^).

#### 3.1.3. EIS Measurements

EIS offers unique information on the electrode reactions and permeability of the surface layer for small ions. The experiments were performed in the equimolar mixture of K_3_[Fe(CN)_6_] and K_4_[Fe(CN)_6_] at the equilibrium potential estimated from cyclic voltammograms. The *R(RC)(RC)* equivalent circuit (5) was used for the data fitting. Here, *R_s_* is the solvent resistance, *R_et_* is the charge transfer resistance, and *CPE* is the constant phase element. The indices ‘1’ and ‘2’ correspond to the inner (electrode–film) and outer (film–solution) interfaces of the electron exchange. The direct current potential for the EIS measurement was determined as an equilibrium potential (a half-sum of the oxidation and reduction peaks of the [Fe(CN)_6_]^3−/4−^). This was equal to 0.229 V for the CB/chitosan coating and to 0.242 V vs. the Ag/AgCl reference electrode for the aptasensor with no respect of the ratio of the other components used in sensor assembly.

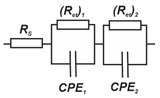
(5)

In the Nyquist diagram (Figure 4a), two semi-circles recorded at high frequencies corresponded to the equilibria of electron transfer on the outer and inner sides of the surface layer. The value of the exponent of the constant phase element (*n*) can be assessed from Equation (6).
(6)$$Z=\frac{1}{{\left(j\omega \right)}^{n}CPE}$$
where *Z* is an impedance; ω is angular frequency of a sinusoidal signal; and *j*^2^ = −1. The *n* value describes non-ideal behavior of the equivalent circuit. Constant phase element has the sense of ideal capacitor for *n* = 1. Here, *n* was found to be between 0.78 and 0.94 for various surface layers due to their roughness and resulted in the formation of slightly flattened semi-circles in the Nyquist diagrams.

The equivalent circuit chosen ascribes the charge transfer conditions that take place on the electrode covered with a rather thick permeable layer so that two independent equillibria exist on the inner and outer interface of the modifying layer. Previously, the formation of two semicircles was observed in impedimetric biosensors utilizing bulky nonconductive but charged molecules like DNA [33,48] or antibodies [49,50], which are able to form differently charged areas in close proximity to the electrode interface. The specific form of the Nyquist diagrams and its relation to the heterogeneity of the surface layer and its properties is discussed in [51,52].

Surprisingly, appropriate charge transfer resistances calculated from the semi-circle diameters were comparable to each other. This contradicts the common behavior of electrodes covered with semipermeable films where the resistance of the inner interface *(R_et_)*_1_ (electrode–film) is much lower than *(R_et_)*_2_ (that of outer side, film–electrolyte solution). Probably this could be attributed to the high accessibility of both sides of the layer to electrolyte ions including ferricyanide ions. The deposition of the OLA-*cone* and aptamer increased the *(R_et_)*_2_, indicating preferable adsorption of the components on the surface of the supporting layer.

To establish specific role of the support components, more traditional self-assembling of polyelectrolyte complexes have also been tested in the aptamer assembly. Here, the OLA-*cone* was deposited directly on GCE and covered with cationic polyelectrolyte PDDA, which worked as a molecular glue and promoted electrostatic adsorption of the aptamer with negative charged phosphate groups of the DNA skeleton. The Nyquist diagrams (Figure 3b) demonstrated predominant changes of the charge transfer resistance on the outer interface and minimal effect of the additives on the conditions of electron transfer on the electrode. The EIS parameters obtained for various surface coatings of the GCE are presented in Table 1.

As can be seen from Table 1, the use of the CB/chitosan support established a low influence of the aptasensor assembly on the conditions of the electron exchange on the electrode surface ((*R_et_*)_1_ values). This might be due to weak changes in the charge of the inner part of the layer. It retains its high permeability toward compact negatively charged [Fe(CN)_6_]^3−/4−^ ions participating in the redox reaction. On the outer interface, deposition of the OLA-*cone*/aptamer exhibited a four-fold increase of the charge transfer resistance due to passive limitation of the redox indicator transfer through the non-conductive components of the layer. Changes in the capacitance were mostly observed in the direction opposite to that of the resistance. This is quite common for impedimetric biosensors and is related to the influence of charge separation on the transfer of the compact anions of redox indicator.

Contrary to that, consecutive deposition of the PDDA and aptamer did not significantly affect the capacitance due to high flexibility of the PDDA molecules and neutralization of its charge by anionic components of the layer. In all cases, the exposition in the KANA solution decreased both *R_et_* values (outer and inner interfaces) and increased the capacitance. This might refer to the conformational changes of the aptamer caused by KANA binding and to the shielding of its negative charge.

Indeed, there are two possible reasons for the *R_et_* changes in impedimetric aptasensors (i.e., (1) passive limitation of the transfer of redox indicator to the electrode, and (2) changes in the electrostatic repulsion between the redox indicator and negatively charged phosphate groups of the DNA aptamer). Here, the decrease of the diffusion rate was disposed by the results of voltammetric study. The shape and position of the redox peaks of indicator remained about the same after the contact with KANA. Thus, shielding of the negative charge remained the only reason for the (*R_et_*)_1_ decrease.

In the case of the PDDA assembly, similar relative shifts of the *R_et_* values were observed at ten-fold higher concentration of the KANA against that used for the aptasensor with the CB/chitosan support (1.0 and 10 nM, respectively). It should be also mentioned that the deposition of the chitosan as a first layer with no CB resulted in unsatisfactory deviation of the EIS parameters, probably due to less effective binding of the aptamers and less reproducibility of the layer composition.

#### 3.1.4. Surface Layer Optimization

Variation of the ratio of the OLA-*cone* and aptamer solutions mixed prior to the deposition on the electrode affected not only the shape of the ferricyanide cyclic voltammogram but also the sensitivity of the response toward KANA exposition. The optimization was conducted using the shift of the *R*_et_ value recorded after 20 min incubation of the aptasensor in 1.0 nM KANA solution. First, it was found that deposition of the aptamer on the CB/chitosan support with no OLA-*cone* did not result in any significant response toward KANA. It is probable that the high porosity of the polymer layer and the charge of the surface caused by the terminal carboxylate groups of the polymer were required for the indication of the KANA binding and vice versa, deposition of the CB/chitosan–OLA-*cone* layer with no aptamer did not respond to the presence of KANA. Thus, all three components (i.e., CB, chitosan, and OLA-*cone*) were necessary for the KANA determination. After that, the ratio of the above-mentioned components was varied from 2:1 to 1:2. The total volume of the aliquot deposited on the GCE was equal to 2 μL in the series of experiments. Regarding the inner interface, the (*R_et_*) value decreased by 30% in the case of the 1:1 ratio of reactants and by 35% for the 2:1 ratio. Absolute shift of the (*R_et_*)_1_ (270 Ω) was higher for the mixture with a bigger relative amount of the aptamer (1:2). At the outer interface (film–solution), the KANA binding shifted the (*R_et_*)_2_ values in the same direction, but to a much lower extent. Relative decay of the resistance was only 13% for the 2:1 ratio and 11% for the 1:2 ratio. It is interesting that the absolute value of the resistance decay was similar on both sides of the surface layer (i.e., about 250–270 Ω). For the 1:1 ratio, no significant changes in the *R_et_* values were found in both cases. Changes in the capacitance followed the shift of the *R_et_* values but were irregular and much less reproducible. The OLA-*cone*/PDDA/aptamer layer responded to 10 nM KANA with 12% increases in the (*R_et_*)_2_ value. Other EIS parameters were about constant. Thus, in the following experiments, the 1:2 ratio of the CB/chitosan and OLA-*cone* was used for KANA determination

The incubation period was another parameter considered in the optimization of the working conditions. The experiments were performed with 10-, 20-, 30-, and 40-min incubation times. To prevent drying, the working surface of the aptasensor was capped with a plastic tube. The incubation was performed at ambient temperature. As shown (see Appendix A), the EIS parameters changed with the incubation period up to 20 min and then remained constant. For this reason, the following determination of the KANA concentration was performed using this incubation period.

### 3.2. KANA Determination

The aptasensor assembled with the 1:2 mixture of the CB/chitosan and the OLA-*cone*/aptamer solutions was incubated in the KANA standard solutions. The KANA binding resulted in regular decrease of the (*R_et_*)_1_ value in the range from 0.7 to 50 nM, then saturation took place (Figure 5).

The appropriate calibration (Equation (7)) is given below.
*R_et_* (Ω) = (707 ± 13) − (199 ± 12) × log(*C*_KANA_, nM), *R*^2^ = 0.9822, n = 6(7)

The calibration was quite stable against the pH variation between 6.5 and 8.0. The limit of detection (LOD) calculated from S/N = 3 ratio was equal to 0.3 nM. This is comparable or better with those reported for electrochemical immuno- and sensors reported for KANA determination. The summary of their characteristics is presented in Table 2. Screen-printed electrodes modified with 4-4-carboxyphenyl radicals followed by covalent attachment of the aptamer were the only direct analog of the developed aptasensor. However, the use of the relative shift of the signal for the KANA determination as well as the necessity of two measurements prior to and after the contact of the aptasensor with the analyte increased the deviation of the response. Besides, both LOD values were significantly lower than those required for milk contamination assessment. The use of the strand displacement offered much lower LODs and KANA concentrations determined. However, extremely high sensitivity was achieved due to multistep amplification including the use of auxiliary DNA strands and DNA polymerization reactions. In the case of the aptasensor array based on solid-state screen-printed electrodes [29], the mechanism of signal generation calls for further investigation. Normally, increased charge of the primary ions decreases the slope of the calibration curve in accordance with the Nernst equation. The potentiometric aptasensor showed the shift of the open circuit potential of about 40 mV per five orders of the magnitude of the KANA concentration. This might be insufficient for reliable measurements. However, even the slope reported seems enormously high due to the multiple charge of the DNA aptamers. It is likely that the dual internal calibration mentioned and exclusion of the reference electrode could provide such characteristics of the aptasensor array.

Changes in the (*R_et_*)_2_ with the KANA concentration were assessed in the same working conditions. They were found to be less regular than (*R_et_*)_1_ though their absolute value was about tenfold higher. It is likely that the KANA molecules also bonded to the aptamer placed on the surface of the OLA-*cone* layer, but not in the pore walls. Such interaction affects the total resistance of the indicator transfer, but tends toward saturation so that the shift of the (*R_et_*)_2_ is observed in a much narrower interval of the KANA concentration and is less reproducible.

### 3.3. Measurement Precision

For estimation of the measurement precision, six individual aptasensors were prepared from the same reagents and applied for the measurement of a 1.0 nM KANA standard solution in PB, pH = 7.0. The sensor-to-sensor repeatability of the signal was found to be 4.8% just after aptasensor preparation and 6.5% after two weeks of their storage at 4 °C. Meanwhile, the average signal did not significantly shift during the whole period of testing.

### 3.4. Real Sample Analysis

The signal of the developed aptasensor was measured in the spiked samples of milk (3.2% fat) and yogurt (2.4% fat) purchased in the local market of Bratislava and Kazan. Prior to measurement, the samples were thermostated at 40 °C for 30 min. Two concentrations (i.e., 30 and 70 nM of KANA) were tested. After addition of the analyte, samples were mixed with methanol in a 1:3 ratio (*v*/*v*) and then centrifuged (see details in the Experimental section). The supernatant was 10-fold diluted with the PB so that the final dilution of the KANA was equal to 30 and the concentration of the KANA was determined at 3.0 and 7.0 nM. The incubation of the aptasensor in diluted milk and yogurt (blank experiment) did not affect the (*R_et_*)_1_ value, but significantly increased the (*R_et_*)_2_ value due to adsorption of the sample components. Nevertheless, the recovery of the KANA determined was quite satisfactory (Table 3). The sensitivity of the aptasensor was sufficient for the reliable detection of the KANA residues at the level of its limited threshold values (about 250 nM). All the aptasensors were used only once to avoid possible errors related to the aging of the surface layer and its contamination with the milk/yogurt components. Attempts to regenerate the aptasensors through treatment with the NaCl solution were unsuccessful. The use of the aptasensors after a long storage period (more than two months) increased deviation of the signal to 10%–12%, but the mean value did not differ significantly from that obtained with the freshly prepared aptasensor.

## 4. Conclusions

In this work, an impedimetric aptasensor was proposed for the determination of KANA by appropriate changes in the permeability of the surfaced layer for ferricyanide redox indicator. The assembly of the surface layer involved casting the CB suspension in chitosan and the addition of the mixture containing the oligolactide derivative with thiacalix[4]arene core in the *cone* conformation and the aptamers. The immobilization of the aptamers was performed by carbodiimide binding. The asymmetrical macrocyclic core resulted in the formation of a branched 3D structure with a high permeability for small ions. This resulted in similar parameters of EIS on both sides of the polymer film. Contrary to similar aptasensors, the KANA binding decreased the charge transfer resistance about threefold in the range from 0.7 to 50 nM. This was attributed to the partial shielding of the negative charge of phosphate residues in the aptamer structure and possible changes in the aggregation of the oligolactide component. Though the linear range of concentrations determined covers 1.5 orders of magnitude, the aptasensor offers some advantages in the determination of the KANA residues in milk and yogurt: the matrix effect was located by changing the EIS parameters on the outer interface of the surface layer whereas the response toward the KANA did not differ dramatically from the results obtained with standard solutions. Thus, the developed aptasensor provides reliable assessment of the KANA contamination within 20 min of incubation and total measurement duration of about 40 min. The recovery of 93%–110% was obtained for the spiked samples of milk and yogurt containing 30 and 70 nM KANA. This content is sufficiently lower than the maximal residue level established in the EU (150 μg/kg, approx. 0.26 μM). The protocol of aptamer immobilization by consecutive accumulation and carbodiimide binding was compared with more conventional electrostatic self-assembling in polyelectrolyte complexes with DPPA. However, the concentrations determined with such a biosensor were ten-fold higher than those with the aptasensor based on the CB/chitosan and OLA-*cone*–aptamer composite. The immobilization protocol elaborated and the phenomenon of decreasing resistance of the inner film–electrode interface can find application in the design of other aptasensors for drug residue contamination detection.

## Figures and Tables

**Figure 1 sensors-20-04738-f001:**
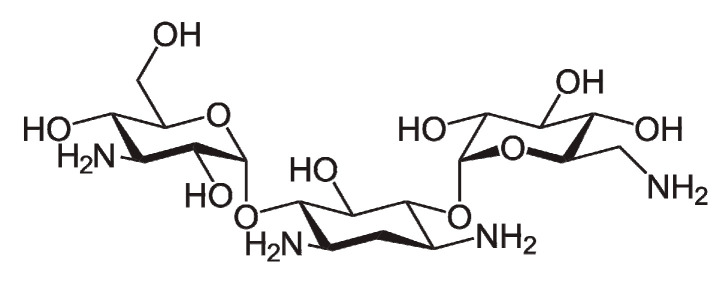
Structural formula of Kanamycin A (KANA).

**Figure 2 sensors-20-04738-f002:**
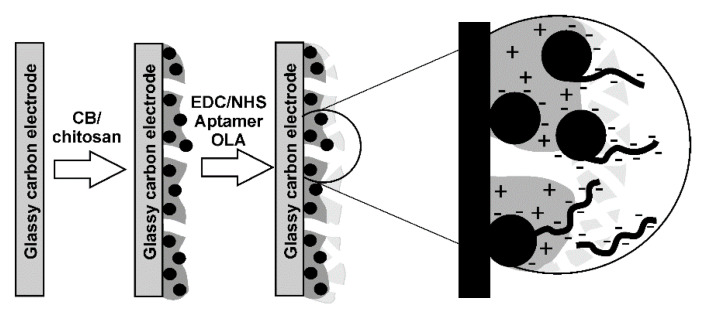
Outline scheme of the aptasensor assembly.

**Figure 3 sensors-20-04738-f003:**
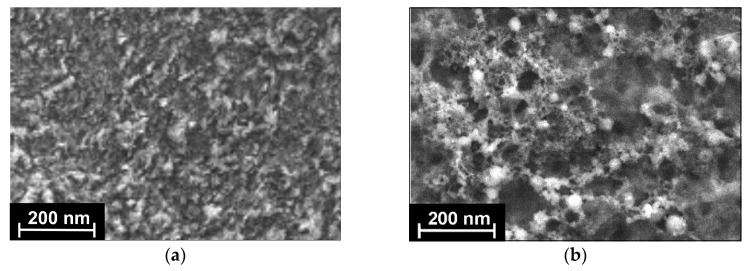
SEM images of the GCE surface covered with the CB cast from a 1.35 mg/mL suspension in 0.1 M HCl containing 0.278% chitosan, 2 μL per electrode (**a**) prior to and (**b**) after deposition of the OLA-*cone*/aptamer mixture (1:2).

**Figure 4 sensors-20-04738-f004:**
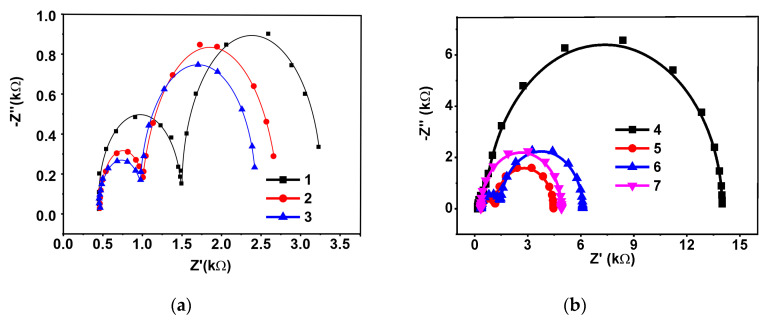
The Nyquist diagram of the impedance spectra recorded on the GCE covered with the CB/chitosan and the OLA-*cone*/aptamer (1:2). (**a**) Immobilization on the CB/chitosan film; (**b**) Immobilization onto PDDA film. Surface layer content: 1-CB/chitosan; 2-CB/chitosan–OLA-*cone*/aptamer (1:2); 3-CB/chitosan–OLA-*cone*/aptamer (1:2)–KANA 1.0 nM; 4-OLA-*cone*; 5-OLA-*cone*–PDDA; 6-OLA-*cone*–PDDA–aptamer; 7-OLA-*cone*–PDDA–aptamer–KANA 10 nM. Measurements were performed in the presence of 0.01 M K_3_[Fe(CN)_6_] and 0.01 M K_4_[Fe(CN)_6_]. Frequency range 0.04 Hz–100 kHz, amplitude 5 mV.

**Figure 5 sensors-20-04738-f005:**
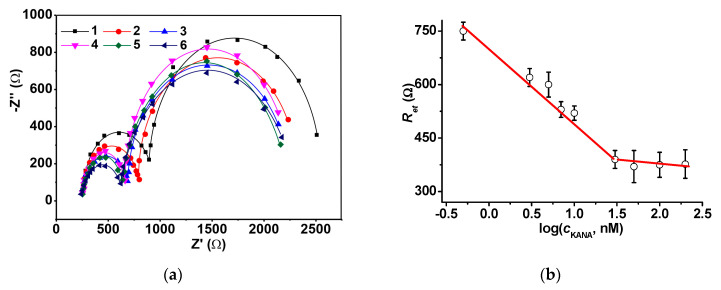
(**a**) The Nyquist diagram of the impedance spectra recorded on the GCE covered with the CB/chitosan and the OLA-*cone*/aptamer mixture (1:2 *v*/*v*) after incubation in 0.5 (1), 3.0 (2), 7.0 (3), 10 (4), 30 (5), and 50 (6) nM KANA; (**b**) The dependence of the (*R_et_*)_1_ value on the KANA concentration. Mean ± S.D. is presented for three individual aptasensors. Measurements in PB, pH = 7.0.

**Table 1 sensors-20-04738-t001:** The EIS parameters of the GCE covered with OLA-*cone*/aptamer (1:2 mixture). Mean ± S.D. for the three individual sensors is presented.

Layer Content	*Rs*(kΩ)	(*R_et_*)_1_(kΩ)	*CPE*_1_(μF)	*n* _1_	(*R_et_*)_2_(kΩ)	*CPE*_2_(μF)	*n* _2_
Aptasensor assembled on the CB/chitosan support							
CB/chitosan	0.45 ± 0.05	0.71 ± 0.10	1.28 ± 0.25	0.94	1.67 ± 0.13	447 ± 10	0.92
+OLA-*cone*/aptamer (1:2)	0.42 ± 0.06	0.60 ± 0.06	1.67 ± 0.30	0.93	6.9 ± 0.3	534 ± 10	0.90
+KANA 1.0 nM (20 min)	0.42 ± 0.07	0.43 ± 0.03	1.41 ± 0.22	0.93	4.3 ± 0.2	556 ± 15	0.90
Aptasensor assembled on the PDDA support							
OLA-*cone*	0.32 ± 0.07	2.9 ± 0.1	6.3 ± 0.1	0.94	12.1 ± 0.8	4.9 ± 0.2	0.90
+PDDA	0.37 ± 0.07	0.64 ± 0.10	0.14 ± 0.03	0.94	5.0 ± 0.1	4.9 ± 0.2	0.85
+KANA 10 nM (20 min)	0.35 ± 0.08	5.7 ± 0.4	0.52± 0.05	0.93	6.9 ± 0.2	4.9 ± 0.2	0.78

**Table 2 sensors-20-04738-t002:** Analytical characteristics of KANA determination with electrochemical immuno- and aptasensors. Fc—ferrocene, MB—Methylene blue.

Modifier	Detection Principle	Concentration Range	LOD	Ref.
Graphene/nanoporous Au/Prussian blue/chitosan, anti-KANA antibodies	Amperometric detection of Prussian blue signal	0.02–14 ng/mL	6.31 pg/mL	[15]
Graphene/nafion/thionine/Pt, anti-KANA antibodies	Amperometric detection of thionine signal	0.01–12 ng/mL	5.74 pg/mL	[16]
Mesoporous Ag@Fe_3_O_4_ nanoparticles/thionine mixed graphene sheets, anti-KANA antibodies	Square wave detection of thionine signal	0.05–16 ng/mL	15 pg/mL	[17]
Multiwalled carbon nanotubes, 1-hexyl-3-methylimidazolium hexafluorophosphate, and nanoporous PtTi, aptamer 5′-NH_2_-AGA TGG GGG TTG AGG CTA AGC CGA-3′	EIS measurements	0.05–100 ng/mL	3.7 pg/mL	[20]
Screen printed carbon electrodes modified by 4-carboxyphenyl, aptamer 5′-TGG GGG TTG AGG CTA AGC CGA-3′-NH_2_	EIS measurements	1.2–600 ng/mL	0.11 ng/mL	[21]
Aptamer-DNA duplex saturated with Methylene blue attached to Au nanoparticles on GCE, aptamer 5′-TGG GGG TTG AGG CTA AGC CGA-3′	Methylene blue mediated horse radish peroxidase reaction, H_2_O_2_ reduction	2.0 pg/mL to 100 ng/mL	0.88 pg/mL	[22]
Ordered mesoporous carbon/chitosan/Au nanoparticles, ferrocene labeled aptamer 5′-ACT TCT CGC AAG ATG GGG GTT GAG GCT AAG CCG AAT ACT CCA GT-Fc-3′)	Strand displacement strategy, ferrocene and ferricyanide signals measured with differential pulse voltammetry	0.1 nM–4.0 μM	0.036 nM (21 pg/mL)	[23]
Pt electrode covered with Ag nanoparticles and covalently attached aptamer 5′-NH_2_-C_6_-AGA TGG GGG TTG AGG CTA AGC CGA-3′	Electroluminescence signal of luminol oxidation	0.5–100 ng/mL	0.06 ng/mL	[24]
Aptamer-DNA duplex with auxiliary strand labeled with ferrocene attached to the Au electrode via terminal thiol group. Capture aptamer: 5′-TGG GGG TTG AGG CTA AGC CGA GTC ACT AT-(CH_2_)_3_-SH	Strand displacement strategy, ferrocene signal measured with square wave voltammetry	1 nM–10 mM	1 nM (0.58 ng/mL)	[25]
Au electrode modified with hairpin aptamer interacting with KANA, two-stage strand displacement and RK polymerase amplification with two auxiliary hairpin DNA sequences. Hairpin aptamer specific to KANA: 5′-TGG GGG TTG AGG CTA AGC CGA CTC AGA GAT CCA TAT GGA ACC CCC A-3′	Measurement of the Methylene blue signal after its intercalation in the polymeric DNA by differential pulse voltammetry	0.05 nM–200 pM	36 fM(0.021 pg/mL)	[26]
Au electrode modified with thiolated aptamer 5′-TGG GGG TTG AGG CTA AGC CGA-3′ hybridized with complementary DNA labeled with Methylene blue	Measurement of the Methylene blue signal after conformational changes of the aptamer bonded to KANA or the shift of labeled DNA closer to the electrode by differential pulse voltammetry	0.2 nM–1.0 μM	0.06 nM (35 pg/mL)	[27]
Au electrode modified with thiolated aptamer 5′-MB-TGG GGG TTG AGG CTA AGC CGA-(CH_2_)_6_-SH-3′ and its lengthened and stem-loop analogs	Measurement of the Methylene blue signal after conformational changes of the aptamer bonded to KANA or the shift of labeled DNA closer to the electrode by differential pulse voltammetry	1.0 nM–100 μM (best characteristics as presented from all the types of aptasensors considered)	0.2 nM (0.11 ng/mL)	[28]
Screen-printed electrode array modified with reduced graphene oxide and Au nanoparticles with attached aptamer 5′-SH-AGA TGG GGG TTG AGG CTA AGC CGA-3′	Measurement of the open circuit potential with dual internal calibration	10 pM–1 μM	5.2 pM (3 pg/mL)	[29]
GCE covered with CB/chitosan—OLA-*cone* and physically adsorbed aptamer	EIS measurements	0.7–50 nM	0.3 Nm (0.17 ng/mL)	This work

**Table 3 sensors-20-04738-t003:** The KANA determination in spiked samples of milk and yogurt. Mean ± S.D. for three individual sensors.

Sample	Added (nM)	Found (nM)	Recovery (%)
Milk (3.2%)	30	33 ± 2	110
	70	74 ± 4	105
Yogurt (2.4%)	30	35 ± 5	117
	70	65 ± 5	93

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
