# Peer review of "Impedimetric Determination of Kanamycin in Milk with Aptasensor Based on Carbon Black-Oligolactide Composite"

_sensors, 2020, doi:10.3390/s20174738_

Round 1

Reviewer 1 Report

The manuscript presents impedimetric determination of Kanamycin with aptasensor based on carbon black-oligolactide composite. Because kanamycin detection strategies with the better detection performance have been already reported (Ref 1-6), this work lacks the novelty enough to be accepted in sensors, Thus, I do not recommend the acceptance of this manuscript:

Ref 1) Sensors and Actuators B: Chemical, 2017, 245, 507-515

Ref 2) Sci Rep. 2017, 7, 40305.

Ref 3) Anal. Chem. 2014, 86, 9372−9375

Ref 4) Sensors and Actuators B: Chemical, 2019, 296, 126664

Ref 5) Anal. Chim. Acta 2018, 1033, 185-192.

Ref 6) Anal. Chim. Acta 2018, 1038, 21-28.

1) The authors need to provide the scheme that clearly shows the construction of proposed sensor.

2) In Fig. 4, the authors need to check the signal at concentrations higher than 50 nM to make sure that it is saturated at concentrations above that.

3) It is hard to find the advantages and novelty of this sensor compared to the above-mentioned papers.

4) In Fig.3 and 4, to compare impedance spectra in Nyquist diagram, all spectra at high frequencies have to start from one point.

5) In Fig. 4, there seems no difference in charge transfer resistance (Rct) of different concentrations (3, 7, 10 and 30 nM), because the start point of the impedance spectra is different.

6) In part 3.1.3 (EIS measurements), the author's description is not convincing. It is better to briefly explain why two semicircles are seen with some references.

7) There are many mistakes. The author needs to check the whole manuscript carefully and try to avoid any grammar or syntax error.

Author Response

We are grateful to the reviewer 1 for useful comments that allowed us to improve the manuscript.

Comment: The manuscript presents impedimetric determination of Kanamycin with aptasensor based on carbon black-oligolactide composite. Because kanamycin detection strategies with the better detection performance have been already reported (Ref 1-6), this work lacks the novelty enough to be accepted in sensors, Thus, I do not recommend the acceptance of this manuscript:

Ref 1) Sensors and Actuators B: Chemical, 2017, 245, 507-515

Ref 2) Sci Rep. 2017, 7, 40305.

Ref 3) Anal. Chem. 2014, 86, 9372−9375

Ref 4) Sensors and Actuators B: Chemical, 2019, 296, 126664

Ref 5) Anal. Chim. Acta 2018, 1033, 185-192.

Ref 6) Anal. Chim. Acta 2018, 1038, 21-28.

Response: We agree that the aptasensor proposed showed not the best characteristics among all the biosensors designed for kanamycin determination. Some of the references kindly provided by Reviewer (Ref. (1), (5) and (6)) have been already mentioned in the manuscript with the following comment: “In most of these cases, additional stages, e.g., substrate addition or secondary DNA strand incubation, are required. Together with necessity of additional modification of aptamers and carrier this complicates both the aptamer assembling and measurement protocol. Besides, many manipulations with biochemical enhancement schemes require high staff qualification and precise conditions of incubation / washing, which can be maintained only in well-equipped laboratory. This limits potential area of application of such biosensors”. The same arguments can be attributed to the Ref. (4). Regarding Refs. (2) (fluorescent detection) and (3) (photoelectrochemical detection), the principles of the signal generation described are far from that used in this work. Optical detection methods have their own drawbacks in the analysis of inhomogeneous samples and mostly assume a number of measurement steps and the use of additional reagents. Besides, optical methods commonly require more sophisticated sample pretreatment to avoid positive false. Thus, only Ref. (1) is indeed a direct analog of the approach proposed. However, relative changes in the EIS parameter are used in this work. They are explained by a large deviation of the background signal and assume hidden doubling of the error (two independent measurements of the signal prior to and after the contact with the sample. From this consideration, the difference in the LOD’s announced (0.11 in ref. (1) vs. 0.17 ng/mL obtained in our work) is not substantially different and does not follow any consequences in the application of the aptasensor for milk analysis where the limited threshold values are much higher than those resulted from the performance of most of the biosensors compared here.

Considering above, the following amendments were made in revised manuscript:

"In the case of fluorescent [30, 31] and photolectrochemical detection [32], the measurement protocol involves several steps with intermediate addition of auxiliary reagents and has some limitations in the analysis of inhomogeneous and turbid samples. Screen-printed electrodes modified with 4-carboxyphenyl radicals followed by covalent attachment of the aptamers is the only direct analog of aptasensor developed. However, the use of relative shift of the signal for the KANA determination as well as necessity in two measurements prior to and after the contact of the aptasensor with the analyte increases the deviation of the response. Besides, both LOD values are significantly lower than those required for milk contamination assessment."

Then, references to Refs. (2) and (3) from the Reviewer’s list were added as [31] and [32].

Comment: 1) The authors need to provide the scheme that clearly shows the construction of proposed sensor.

Response: The Figure 1 was added in revised manuscript that shows the construction of the sensor.

Comment: 2) In Fig. 4, the authors need to check the signal at concentrations higher than 50 nM to make sure that it is saturated at concentrations above that.

Response: Fig.4b was re-plotted to show the signal at high KANA concentrations.  

Comment: 3) It is hard to find the advantages and novelty of this sensor compared to the above-mentioned papers.

Response: The following text was added to the last paragraph of the Introduction section.

"For the first time, the composite consisted of the CB, chitosan and thiacalix[4]arene bearing oligolactide fragments has been used as support for the aptamer immobilization. The formation of highly porous layer made it possible to increase the quantities of the aptamer attained in the layer and preserve its high permeability for small molecules. The interaction with the analytes resulted in decrease of the charge transfer resistance on the inner interface of the layer. To the best of our knowledge, only increased resistance was previously observed in label free aptasensors. The peculiarities of the surface layer structure and of the signal transduction made the aptasensor less sensitive toward matrix compounds of the samples tested."

Comment: 4) In Fig.3 and 4, to compare impedance spectra in Nyquist diagram, all spectra at high frequencies have to start from one point.

5) In Fig. 4, there seems no difference in charge transfer resistance (Rct) of different concentrations (3, 7, 10 and 30 nM), because the start point of the impedance spectra is different.

Response: The EIS data were reconsidered and revised Figures concerned are presented in revised manuscript. As it could be seen, the charge transfer resistance corresponded to the inner interface (smaller semi-circle in Fig.4) changes with the KANA concentration in accordance with the calibration curve (Fig. 4b).

Comment: 6) In part 3.1.3 (EIS measurements), the author's description is not convincing. It is better to briefly explain why two semicircles are seen with some references.

Response: The following text was added to the Section 3.1.3 of revised manuscript:

"The equivalent circuit chosen ascribes the charge transfer conditions that take place on the electrode covered with rather thick permeable layer so that two independent equillibria exist on inner and outer interface of the modifying layer. Previously, the formation of two semicircles was observed in impedimetric biosensors utilizing bulky nonconductive but charged molecules like DNA [34, 44] or antibodies [49, 50] that are able to form differently charged areas in the close proximity to the electrode interface. The specific form of the Nyquist diagrams and its relation to the heterogeneity of the surface layer and its properties is discussed in [51, 52]."

New references [49-52] were added in reference list of revised manuscript.

Comment: 7) There are many mistakes. The author needs to check the whole manuscript carefully and try to avoid any grammar or syntax error.

Response: We have carefully checked the manuscript to avoid misprints and syntax errors.

Reviewer 2 Report

The authors report on an impedimetric aptasensor modified with carbon black for the detection of Kanamycin in milk. The innovation of this article is based on the addition of carbon black that is related to an improvement in the sensitivity. However, to confirm this hypothesis a comparison of the platforms is required. Moreover, the discussion of EIS results must be revised and improved. Below other points that must be revised.

Please carefully check for typographical errors in the text. e.g. page 4.

Please explain the rationale to add the carbon black in the aptasensor architecture. Wasn’t the previous platform sensitive enough?

I suggest you add an easily understandable illustration to explain aptasensor fabrication and the sensing mechanism.

Please describe in the text the DC potential applied in the EIS measurements.

Please present the voltammetric response of all mixtures in the Support information.

Please clarify the reason to present in Fig. 1a the voltammetric response of film architecture 1:1, and in the Fig. 1b the response of 1:2. Why the response of 2:1 was not presented in eq. 2?

Please explain why the diffusion process is not observed in EIS experiments since frequency scanning was up to 0.04 Hz.

Please present references that explain the existence of the two semicircles, because it is not usual relate the response of the film-solution. From SEM images it seems that the film does not cover all the electrode surface, most probably the EIS response may come from this heterogeneous surface.

Please change the constant phase element abbreviation from C to CPE. It may confuse the reader. Moreover, the table must include the Rsol and n values.

Nyquist diagram are usually presented in squared plots.

The solution resistance is usually constant for all measurements. Please explain the shift observed in Fig. 3a and 4a.

Real sample response also must be included in the manuscript or in the Support Information.

Please explain how the detection limit was calculated and present the analytical parameters obtained from linear fit of Fig. 4b.

Author Response

We are grateful to reviewer 2 for useful comments that allowed us to improve the manuscript.

The authors report on an impedimetric aptasensor modified with carbon black for the detection of Kanamycin in milk. The innovation of this article is based on the addition of carbon black that is related to an improvement in the sensitivity. However, to confirm this hypothesis a comparison of the platforms is required. Moreover, the discussion of EIS results must be revised and improved. Below other points that must be revised.

Comment: Please explain the rationale to add the carbon black in the aptasensor architecture. Wasn’t the previous platform sensitive enough?

Response: Indeed, the novelty and advantage of the research is directed to the whole content of the support providing the formation of highly permeable layer with an excess of small reactants (KANA and redox probe ions) to the inner volume of the layer. The function of the CB is rather common and consists of two aspects, i.e., increased conductivity of the matrix and support for covalent attachment of the aptamers. The necessity of the layer components was proved by comparison of the results with those obtained for layer-by-layer composition of the surface layer. We have described the novelty of our response in more detail in revised manuscript as follows:

"For the first time, the composite consisted of the CB, chitosan and thiacalix[4]arene bearing oligolactide fragments has been used as support for the aptamer immobilization. The formation of highly porous layer made it possible to increase the quantities of the aptamer attained in the layer and preserve its high permeability for small molecules. The interaction with the analytes resulted in decrease of the charge transfer resistance on the inner interface of the layer. To the best of our knowledge, only increased resistance was previously observed in label free aptasensors. The peculiarities of the surface layer structure and of the signal transduction made the aptasensor less sensitive toward matrix compounds of the samples tested."

Besides, previous experience of the CB application described elsewhere was mentioned already in the Introduction (refs. [30-36]). The necessity in the CB was also proved by our own experiments not mentioned in the previous version of the manuscript due to negative results. Now, the following text was added at the end of the Section 3.1.3:

"It should be also mentioned that the deposition of the chitosan as first layer with no CB resulted in unsatisfactory deviation of the EIS parameters probably due to less effective binding of aptamers and less reproducibility of the layer composition."

Comment: Please carefully check for typographical errors in the text. e.g. page 4.

Response: We have carefully checked the manuscript to avoid misprints and syntax errors.

Comment: I suggest you add an easily understandable illustration to explain aptasensor fabrication and the sensing mechanism.

Response: We added Figure 1 that explains the fabrication of the aptasensor.

Comment: Please describe in the text the DC potential applied in the EIS measurements.

Response: The following text was added to the Section 3.1.3 prior to Eq. (6):

"The direct current potential for the EIS measurement was determined as equilibrium potential (a half sum of the oxidation and reduction peaks of the [Fe(CN)6]3-/4-). It was equal to 0.229 V for the CB/chitosan coating and to 0.242 V vs. Ag/AgCl reference electrode for the aptasensor with no respect of the ratio of the other components used in sensor assembling."

Comment: Please present the voltammetric response of all mixtures in the Support information.

Response: As requested, the cyclic voltammograms have been moved to the Supplementary material as Figure S1.

Comment: Please clarify the reason to present in Fig. 1a the voltammetric response of film architecture 1:1, and in the Fig. 1b the response of 1:2.

Response: The cyclic voltammograms obtained for the 2:1 mixture were very similar to those presented for 1:1 mixture. This was mentioned in the text at page 7 as follows:

"As could be seen from the equations (2), the conditions for the transfer of the redox indicator were about similar for the 2:1 and 1:1 mixture though 1:2 mixture indicated increased diffusional resistance of the transfer."

Comment: Why the response of 2:1 was not presented in eq. 2?

Response: The response of 2:1 was added as required.

Comment: Please explain why the diffusion process is not observed in EIS experiments since frequency scanning was up to 0.04 Hz. Please present references that explain the existence of the two semicircles, because it is not usual relate the response of the film-solution. From SEM images it seems that the film does not cover all the electrode surface, most probably the EIS response may come from this heterogeneous surface.

Response: Similar Nyquist diagrams were reported for biosensors with highly porous and rather thick layers where the charge transfer is located within the layer so that diffusional transfer of the redox probe is not taken into account. The shape of the curves on Fig. 3 was described as follows (the text is added to the Section 3.1.3) as follows:

"The equivalent circuit chosen ascribes the charge transfer conditions that take place on the electrode covered with rather thick permeable layer so that two independent equillibria exist on inner and outer interface of the modifying layer. Previously, the formation of two semicircles was observed in impedimetric biosensors utilizing bulky nonconductive but charged molecules like DNA [29, 39] or antibodies [49, 50] that are able to form differently charged areas in the close proximity to the electrode interface. The specific form of the Nyquist diagrams and its relation to the heterogeneity of the surface layer and its properties is discussed in [51, 52]."

New references [49-52] were added in reference list of revised manuscript.

We would like also to draw the attention of esteemed Reviewer, that the equivalent circuit chosen (Eq. 6) does not contain the Warburg impedance as recommended in the Ref. 46. In Ref. [50] with the same equivalent circuit, no diffusion related piece of curve was observed on the Nyquist diagram though the frequency range was similar to that used in this work (0.1 Hz – 100 kHz).

Comment: Please change the constant phase element abbreviation from C to CPE. It may confuse the reader. Moreover, the table must include the Rsol and n values. Nyquist diagram are usually presented in squared plots.

Response: The required changes and amendments have been made.

Comment: The solution resistance is usually constant for all measurements. Please explain the shift observed in Fig. 3a and 4a.

Response: We apologize for technical mistake with the EIS data presentation. The results were reconsidered and the graphs at Figures 3a and 4a were replotted accordingly.             

Comment: Please explain how the detection limit was calculated and present the analytical parameters obtained from linear fit of Fig. 4b.

Response: The calculation of the LOD was explained directly after the Figure 4 (S/N = 3).  Other characteristics followed from the linear fit were presented prior to the Figure (Eq. (8) and the text below).

Reviewer 3 Report

The authors presented an impedimetric aptasensor for the determination of KANA by changing the permeability of the surfaced layer for ferricyanide redox indicator. The sensor is composed by a surface layer of CB casted in chitosan and added with a mixture containing oligolactide derivative with thiacalix[4]arene core in cone conformation. They immobilised specific aptamers by carbodiimide crosslink chemistry.

The work is well organized and the experiments, results and discussion reported are satisfactory. 
Overall, the reviewer think that this work is significant and can be publish on Sensor after few minor revision.

Here some suggestion to facilitate the final reader:

  • Provide a schematic of the overall architecture and strategy of the sensor
  • To help the reader, I suggest to implement the manuscript with a supplementary information file where you add results discussed in the main text. For example for stamens I line 198-199 "Other parameters, i.e., pH of the buffer solution or electrolyte content did not affect deposition of the CB on the GCE" and statement in line 211-215 
  • please provide schematic for OLA-cone if possible
  • In figure 2 it would be better to compare images with same magnification
  • In paragraph "3.3. Measurement precision" authors report an increase of almost 2% of the signals after 2 weeks of storage at 4°C, please provide some speculation on the reason why. It would be also helpful to provide the graph with results in the supplementary information.
  • Very few typos are present in the text, please check

Author Response

We are grateful to reviewer 3 for useful comments that allowed us to improve the manuscript

The authors presented an impedimetric aptasensor for the determination of KANA by changing the permeability of the surfaced layer for ferricyanide redox indicator. The sensor is composed by a surface layer of CB casted in chitosan and added with a mixture containing oligolactide derivative with thiacalix[4]arene core in cone conformation. They immobilised specific aptamers by carbodiimide crosslink chemistry. The work is well organized and the experiments, results and discussion reported are satisfactory. Overall, the reviewer think that this work is significant and can be publish on Sensor after few minor revisions.

Here some suggestion to facilitate the final reader:

Comment: Provide a schematic of the overall architecture and strategy of the sensor.

Response: The new Figure 1 with the requested architecture of the sensor was included in revised manuscript.

Comment: To help the reader, I suggest to implement the manuscript with a supplementary information file where you add results discussed in the main text. For example for stamens I line 198-199 "Other parameters, i.e., pH of the buffer solution or electrolyte content did not affect deposition of the CB on the GCE" and statement in line 211-215. 

Response: The information on the influence of the pH and phosphate buffer concentration on the CB deposition was added to the Supplementary material (Table S1). The surface concentration of the aptamer was added to the text.

Comment: Please provide schematic for OLA-cone if possible

Response: This structure was added to the Supplementary material and Scheme 1.

Comment: In figure 2 it would be better to compare images with same magnification

Response: The Figures 2a and 2b are presented in revised manuscript in the same magnification as requested.

Comment: In paragraph "3.3. Measurement precision" authors report an increase of almost 2% of the signals after 2 weeks of storage at 4°C, please provide some speculation on the reason why. It would be also helpful to provide the graph with results in the supplementary information.

Response: We apologize for inconvenience with the data presented. Indeed, the deviation of the signals was increased from 4.8 to 6.5%. The mean value remains the same for the whole testing period (two weeks). This information was added to the revised text.

Comment: Very few typos are present in the text, please check

Response: We have carefully checked the manuscript to avoid misprints and syntax errors.

Reviewer 4 Report

 Kulikova et al.  developed an impedimetric aptasensor to determine KANA in milk and yogurt, which showed a promising sensitivity and accuracy. However, there are some issues that need to be addressed.

  1. The introduction is not well-written; the following are some examples:
  • Line 68: “by combinatorial chemistry approach and then selected against an analyte by affine chromatography” should be … affinity chromatography.
  • Line 69:” They are considered as an alternative to traditional antibodies due to rather simple manufacture, …” this sentence is not clear and needs to be written.
  • Line 77-79: Sentence is too long and it’s hard to understand.
  • Line 83- 86: Sentence is too long and it’s hard to understand.

I strongly suggest that authors rewrite and reorganize the introduction.

  1. Also, there are a lack of recent publication related to the topic, which needs to be discussed. Authors should discuss how their biosensor stands out from the similar aptamer-based sensors that were recently published. These are a few examples of recent publications:
  • Han, X. et al. (2019). Two kanamycin electrochemical aptamer-based sensors using different signal transduction mechanisms: A comparison of electrochemical behavior and sensing performance. Bioelectrochemistry, 129, 270-277. doi:10.1016/j.bioelechem.2019.06.004
  • Jiang, Y. et al. (2020). A simple and sensitive aptasensor based on SERS for trace analysis of kanamycin in milk.Journal of Food Measurement and Characterization, doi:10.1007/s11694-020-00553-7
  • Wang, J. et al. (2020). A label-free and carbon dots based fluorescent aptasensor for the detection of kanamycin in milk.Spectrochimica Acta - Part A: Molecular and Biomolecular Spectroscopy, 226 doi:10.1016/j.saa.2019.117651
  • Yu, Z. et al. (2020). Lengthening the aptamer to hybridize with a stem-loop DNA assistant probe for the electrochemical detection of kanamycin with improved sensitivity.Analytical and Bioanalytical Chemistry, 412(11), 2391-2397. doi:10.1007/s00216-020-02481-3
  • Yu, J. et al. (2020). Simultaneous detection of streptomycin and kanamycin based on an all-solid-state potentiometric aptasensor array with a dual-internal calibration system.Sensors and Actuators, B: Chemical, 311 doi:10.1016/j.snb.2020.127857

  1. Line 107-109: Please provide the appropriate reference for the synthesis of carboxylated thiacalix[4]arene and OLA-cone (Reference 35 is not related to the topic) or explain the procedure in details.
  2. Line 121-124: pH of the buffer and the supporting electrolyte needs to be mentioned for the electrochemistry measurements.
  3. Line 139: How many cycles were used for electrochemical cleaning? Also, the electrochemical technique that was used needs to be mentioned.
  4. Line 148-149: What was the shelf-time of the aptasensor?
  5. Line 172-173: Please provide the scan rate plot.
  6. Line 174-178: the discussion is not clear: Figure 1b shows the CV at different scan rates, which is not related to the statement.
  7. Line 179-182: Authors claimed that “Incubation in KANA solution did not alter the peak potentials but increased appropriate currents.” However, there is a slit shift in the peak potential, which should be discussed accordingly.
  8. Line 200-204: Why for this experiment [Ru(NH3)6]3+ redox probe was used not Fe3+/4+? Authors should provide reason and justify their decision.
  9. Line 213: Please provide the value for the Γ Apt.
  10. Line 213-214: Authors claimed that “ The experiments confirm the immobilization of the aptamers and their accessibility for small charge carriers in the polymer matrix.” This claim needs to be explained in more detail.
  11. Figure 2: Please provide a higher resolution SEM images with the same scale.
  12. Line 223-224: Authors claimed, “ The addition of the OLA-cone / aptamer mixture resulted in formation of hierarchical 3D structure with rather big and tangled channels.” What is the evidence for this claim?
  13. Line 228-230: Authors mentioned, “Highly porous layer can be also formed due asymmetrical structure of the thiacalix[4]arene in cone macrocyclic core with all the elongated substituents positioned from one side of the macrocycle.” This needs to be explained with supporting evidence.
  14. Line 238-239: “On the Nyquist diagram (Fig. 3(a)), two semi-circles recorded at high frequencies correspond to the equilibria of electron transfer on the outer and inner sides of the surface layer.” Observation of two semicircles in Nyquist plot is interesting, but it is not common for electrochemical sensors. Authors should discuss this more comprehensively by comparing with similar observation. Also, the claim needs to be justified by the evidence.
  15. Table 1: The trends of Ret 1 and Ret 2 should be discussed in detail.
  16. Line 279-280: “This might be referred to the conformational changes of the aptamer caused by KANA binding and shielding its negative charge.” The authors should provide supporting evidence for their claim.
  17. Line 296-297: “Regarding inner interface, the (Ret) value decreased by 30% in case of 2:1 ratio of reactants and by 35% for 2:1 ratio.” The ratio needs to be corrected in the above sentence.
  18. Line 307-312: Please provide a table for the time-dependent experiment.
  19. Figure 4b: Why only graph for (Ret) 1 was provided and discussed?
  20. Line 378: “Thus, the aptasensor developed ….” Should be “the developed aptasensor”
  21. Line 382-384: Please provide data and/or calibration curve for OLA-cone-aptamer composite modification.

Author Response

We are grateful to reviewer 4 for useful comments that allowed us to improve the manuscript.

Kulikova et al.  developed an impedimetric aptasensor to determine KANA in milk and yogurt, which showed a promising sensitivity and accuracy. However, there are some issues that need to be addressed.

Comment 1: The introduction is not well-written; the following are some examples:

Line 68: “by combinatorial chemistry approach and then selected against an analyte by affine chromatography” should be … affinity chromatography.

Line 69:” They are considered as an alternative to traditional antibodies due to rather simple manufacture, …” this sentence is not clear and needs to be written.

Line 77-79: Sentence is too long and it’s hard to understand.

Line 83- 86: Sentence is too long and it’s hard to understand.

I strongly suggest that authors rewrite and reorganize the introduction.

Response: We have re-written the Introduction. All of the above requirements of the reviewer have been taken into account.

Comment: Also, there are a lack of recent publication related to the topic, which needs to be discussed. Authors should discuss how their biosensor stands out from the similar aptamer-based sensors that were recently published. These are a few examples of recent publications:

Han, X. et al. (2019). Two kanamycin electrochemical aptamer-based sensors using different signal transduction mechanisms: A comparison of electrochemical behavior and sensing performance. Bioelectrochemistry, 129, 270-277. doi:10.1016/j.bioelechem.2019.06.004

Jiang, Y. et al. (2020). A simple and sensitive aptasensor based on SERS for trace analysis of kanamycin in milk. Journal of Food Measurement and Characterization, doi:10.1007/s11694-020-00553-7

Wang, J. et al. (2020). A label-free and carbon dots based fluorescent aptasensor for the detection of kanamycin in milk. Spectrochimica Acta - Part A: Molecular and Biomolecular Spectroscopy, 226 doi:10.1016/j.saa.2019.117651

Yu, Z. et al. (2020). Lengthening the aptamer to hybridize with a stem-loop DNA assistant probe for the electrochemical detection of kanamycin with improved sensitivity. Analytical and Bioanalytical Chemistry, 412(11), 2391-2397. doi:10.1007/s00216-020-02481-3

Yu, J. et al. (2020). Simultaneous detection of streptomycin and kanamycin based on an all-solid-state potentiometric aptasensor array with a dual-internal calibration system. Sensors and Actuators, B: Chemical, 311 doi:10.1016/j.snb.2020.127857

Response: Fluorescent and SERS biosensors differ from electrochemical biosensors with no respect of the analyte nature and mostly show better analytical characteristics. Meanwhile they show some drawbacks like more complicated design, necessity of expensive modification on biochemical reagents, multistep assay protocol and some limitations in the analysis of nonhomogeneous and turbid samples. These arguments have been included in the Introduction section. Other references from the above list were included in the Table 2 and briefly described in Introduction.

Comment: Line 107-109: Please provide the appropriate reference for the synthesis of carboxylated thiacalix[4]arene and OLA-cone (Reference 35 is not related to the topic) or explain the procedure in details.

Response: The following text was added to the Section 2.1:

"Thiacalix[4]arene in cone configuration bearing oligolactide fragments and denoted as OLA-cone was synthesized at the Organic Chemistry Department of Kazan Federal University. Briefly, 5,11,17,23-tetra-tert-butyl- 25,26,27,28-tetrakis[(hydroxycarbonyl) methoxy]-thiacalix[4]arene in cone configuration was first synthesized as described in [41]. Oligolactic acid was obtained by heating lactic acid in rotary evaporator, 270 rpm, at 180 oC and 20-40 mbar. Then the reactants were mixed and heated in the argon atmosphere for four hours in the presence of 0.01% Sn dioctate as catalyst.  It provided extension of the oligolactide fragments to 13-17 monomer units from 5-6 fragments obtained in reaction of the carboxylated thiaclix[4]arene and lactic acid. The structure of the product was earlier confirmed by the NMR 1H spectra and MALDI TOF spectroscopy [42]. The reaction scheme is presented in Supporting Information as Scheme S1."

[42] Kuzin, Yu.I.; Gorbatchuk V.V.; Rogov, A.M.; Stoikov, I.I.; Evtugyn, G.A. Electrochemical properties of multilayered coatings implementing thiacalix[4]arenes with oligolactic fragments and DNA. Electroanalysis 2020, 32, 715-723. DOI: 10.1002/elan.201900499

The ref.[38] describes the application of oligolactide in the area of electrochemical DNA sensors and meets the topic of the work.

Comment: Line 121-124: pH of the buffer and the supporting electrolyte needs to be mentioned for the electrochemistry measurements.

Response: The following text was introduced in the revised manuscript, section 2.1:

"Electrochemical measurements were performed at pH = 7.0 in 25 mM phosphate buffer (PB) consisted of 24.7 mM Na2HPO4 and 0.3 mM NaH2PO4. The pH was adjusted by 1.0 M NaOH and 1.0 M HCl. As a supporting electrolyte 0.1 M KCl was used."

Comment: Line 139: How many cycles were used for electrochemical cleaning? Also, the electrochemical technique that was used needs to be mentioned.

Response: The following sentence was introduced in section 2.3. as requested:

"10-15 cycles were recorded in DC mode until stabilization of the voltammograms."

Comment: Line 148-149: What was the shelf-time of the aptasensor?

Response: The following sentence was included in the section 2.3:

"The developed aptasensors retained their main operation characteristics within two months storage."

It should be also noted that changes in the measurement repeatability are briefly mentioned later on in the Section 3.4 and in Conclusion.

Comment: Line 172-173: Please provide the scan rate plot.

Response: The voltammetric measurements are not used for the KANA determination and only show permeability of the surface layer toward small ions necessary for EIS measurements and KANA binding observation. For this reason, Reviewer 2 recommended to move the voltammograms To Supplementary material. The appropriate equations (Eq.2) indicated satisfactory linear fitting of the scan rate plots for all the sensor layers. It seems sufficient for this step of the surface layer characterization. Nevertheless, we have added the scan rate plot for as Figure S2 to Supplementary material.

Comment: Line 174-178: the discussion is not clear: Figure 1b shows the CV at different scan rates, which is not related to the statement.

Response: The reference to the Fig. 1b was removed and Figure 1 was moved to the Supplementary material because of the requirement of the Reviewer 2. Instead, it is suggested to compare the Figs. S1(a) and S1(b).

Comment: Line 179-182: Authors claimed that “Incubation in KANA solution did not alter the peak potentials but increased appropriate currents.” However, there is a slit shift in the peak potential, which should be discussed accordingly.

Response: The shift in the peak potential was observed after the deposition of the OLA-cone/aptamer layer but not after the incubation in the KANA solution. Changes in the equilibrium potentials were explained as follows in revised manuscript (Section 3.1.1):

Increased quantities of the aptamers changed the peak due to higher content of the nonconductive components in the surface layer (compare the Figs. S1(a) and S1(b)).

Comment: Line 200-204: Why for this experiment [Ru(NH3)6]3+ redox probe was used not Fe3+/4+? Authors should provide reason and justify their decision.

Response: We used standard protocol described elsewhere (Steel, A.B.; Herne, T.M.; Tarlov, M.J. Electrochemical quantitation of DNA immobilized on gold. Anal. Chem. 1998, 70, 4670-4677. DOI: 10.1021/ac980037q). It is commonly used for the assessment of the DNA concentration of the electrode surface. The reference to the method was given in the text [47]. Contrary to [Fe(CN)6]3-/4-, which is negatively charged like DNA bearing phosphate groups, Ru complex is a cation that attaches to the nucleotide and preserves its redox activity in the complex. This allows assessing the DNA concentration by measurement of the charge transferred in redox conversion of the Ru complex after transferring the aptasensor to the solution with no indicator. Some explanations are given as follows to the text (Section 3.1.1):

"Contrary to [Fe(CN)6]3-/4-, Ru complex reacts quantitatively with the DNA nucleobases and remains electrochemically active after accumulation in the aptamer layer. After that, the aptasensor was transferred into the PB with no indicator and the charge transferred in the redox conversion of [Ru(NH3)6]3+ was measured."

Comment: Line 213: Please provide the value for the Γ Apt.

Response: Requested values were added as follows (Section 3.1.1):

"Changes in the quantities of the aptamers by utilizing other mixtures (2:1, 1:1) gave correspondingly changing values of the  ((5.0 ± 0.5) × 1013, and (6.2 ± 0.5) × 1013 respectively). Thus, the experiments with [Ru(NH3)6]3+ redox indicator confirmed the attachment of the aptamer to the underlying layer and the voltammetric experiments with [Fe(CN)6]3-/4- revealed high permeability of the surface layer for small ions."

Comment: Line 213-214: Authors claimed that “The experiments confirm the immobilization of the aptamers and their accessibility for small charge carriers in the polymer matrix.” This claim needs to be explained in more detail.

Response: Actually, this statement just repeats intermediate conclusions that were above introduced in the text of the Section 3.1.1. The shape of the [Fe (CN)6]3-/4- peaks and their changes within the surface layer assembling confirm permeability of the surface layer for small ions. The experiments with [Ru(NH3)6]3+ redox indicator described just before this phrase explicitly confirm the DNA aptamer immobilization. We suggest only to re-phrase the sentence as follows (Section 3.1.1):

"Thus, the experiments with [Ru(NH3)6]3+ redox indicator confirmed the attachment of the aptamer to the underlying layer and the voltammetric experiments with [Fe(CN)6]3-/4- high permeability of the surface layer for small ions."

Comment: Figure 2: Please provide a higher resolution SEM images with the same scale.

Response: Figure 2a was presented with the same resolution as Figure 2b.

Comment: Line 223-224: Authors claimed, “The addition of the OLA-cone / aptamer mixture resulted in formation of hierarchical 3D structure with rather big and tangled channels.” What is the evidence for this claim?

Line 228-230: Authors mentioned, “Highly porous layer can be also formed due asymmetrical structure of the thiacalix[4]arene in cone macrocyclic core with all the elongated substituents positioned from one side of the macrocycle.” This needs to be explained with supporting evidence.

Response: We believe that the high permeability of the covering layer for ferricyanide ions even though they are negatively charged like terminal carboxylic groups of OLA-cone layer together with SEM image presented testify in favor of such hypothesis. To avoid confusing we suggest to remove the word ‘hierarchical’ and introduce ‘highly porous’ instead. The sentence from the line 228-230 was re-phrased and moved back. Thus, the following changes are proposed for the paragraph discussed (Section 3.1.2):

"The addition of the OLA-cone / aptamer mixture resulted in formation of highly porous 3D structure with rather big and tangled channels. This coincides well with similar investigations of the oligolactide layer after cathodic deposition of elemental silver: metal particles formed dendritic palm tree like structures with a high number of branching and a high variation of dimensions [48]. The formation of silver dendrites was affected by asymmetrical structure of the thiacalix[4]arene in cone macrocyclic core with all the substituents positioned from one side of the macrocycle core. Such a structure of the aptamer layer explains low level of diffusional limitations observed on cyclic voltammograms of the redox indicator ([Fe(CN)6]3-/4-)."

Comment: Line 238-239: “On the Nyquist diagram (Fig. 3(a)), two semi-circles recorded at high frequencies correspond to the equilibria of electron transfer on the outer and inner sides of the surface layer.” Observation of two semicircles in Nyquist plot is interesting, but it is not common for electrochemical sensors. Authors should discuss this more comprehensively by comparing with similar observation. Also, the claim needs to be justified by the evidence.

Response: The following text was introduced to explain the EIS data (Section 3.1.3):

"The equivalent circuit chosen ascribes the charge transfer conditions that take place on the electrode covered with rather thick permeable layer so that two independent equillibria exist on inner and outer interface of the modifying layer. Previously, the formation of two semicircles was observed in impedimetric biosensors utilizing bulky nonconductive but charged molecules like DNA [29, 39] or antibodies [49, 50] that are able to form differently charged areas in the close proximity to the electrode interface. The specific form of the Nyquist diagrams and its relation to the heterogeneity of the surface layer and its properties is discussed in [51, 52]."

New references [49-52] were added in reference list of revised manuscript.

Comment: Table 1: The trends of Ret 1 and Ret 2 should be discussed in detail.

Response: The following paragraph was introduced in the text after Table 1: 

"As could be seen from Table 1, the use of the CB/chitosan support establishes low influence of the aptasensor assembling on the conditions of the electron exchange on the electrode surface ((Ret)1 values). This might be due to weak changes in the charge of the inner part of the layer. It retains its high permeability toward compact negatively charged [Fe(CN)6]3-/4- ions participating in the redox reaction. On the outer interface, deposition of the OLA-cone/aptamer exhibited four-fold increase of the charge transfer resistance due to passive limitation of the redox indicator transfer through the non-conductive components of the layer. Changes in the capacitance are mostly observed in the direction opposite to that of the resistance. This is quite common for impedimetric biosensors and is related to the influence of charge separation on the transfer of compact anions of redox indicator."

Comment: Line 279-280: “This might be referred to the conformational changes of the aptamer caused by KANA binding and shielding its negative charge.” The authors should provide supporting evidence for their claim.

Response: In EIS experiments, all the changes in the charge transfer resistance measured in the presence of the ferricyanide ions as redox indicators are mostly explained by two reasons: (1) an analyte binding decreases the diffusion rate of the indicator (passive limitation of the indicator transfer) and (2) the charge transfer resistance increases due to the electrostatic repulsion of negatively charged ferricyanide ions from negatively charged phosphate residues of the DNA backbone. In our case, passive limitation of the indicator transfer was declined due to results of voltammetric experiment with ferricyanide ion (position and shape of the redox peaks did not change dramatically after the KANA binding). Thus, only charge shielding remained as possible reason of the (Ret)1 decrease. The following paragraph was introduced in the section 3.1.3:

"Indeed, there are two possible reasons for the Ret changes in impedimetric aptasensors, i.e. (1) passive limitation of the transfer of redox indicator to the electrode, and (2) changes in the electrostatic repulsion between the redox indicator and negatively charged phosphate groups of the DNA aptamer. Here, the decrease of the diffusion rate was disposed by the results of voltammetric study. The shape and position of the redox peaks of indicator remained about the same after the contact with KANA. Thus, shielding of the negative chagre remained the only reason of the (Ret)1 decrease."

Comment: Line 296-297: “Regarding inner interface, the (Ret) value decreased by 30% in case of 2:1 ratio of reactants and by 35% for 2:1 ratio.” The ratio needs to be corrected in the above sentence.

Response: We apologize for technical mistake, the ratio mentioned was corrected.

Comment: Line 307-312: Please provide a table for the time-dependent experiment.

Response: Table S2 with time-dependent experiment results was added to the Supplementary material.

Comment: Figure 4b: Why only graph for (Ret) 1 was provided and discussed?

Response: The following text was added in section 3.2:

"Changes in the (Ret)2 with the KANA concentration were assessed in the same working conditions. They were found to be less regular than (Ret)1 though their absolute value was about tenfold higher. Probably, the KANA molecules are also bonded to the aptamer placed on the surface of the OLA-cone layer but not in the pore walls. Such interaction affects the total resistance of the indicator transfer but tends to saturation so that the shift of the (Ret)2 is observed in much narrower interval of the KANA concentration and is less reproducible."

Comment: Line 378: “Thus, the aptasensor developed ….” Should be “the developed aptasensor”

Response: We have changed the word order as requested.

Comment: Line 382-384: Please provide data and/or calibration curve for OLA-cone-aptamer composite modification.

Response: The following sentences were added in Conclusion:

"The recovery of 93-110% was obtained for the spiked samples of milk and yogurt containing 30 and 70 nM KANA. This content is sufficiently lower than the maximal residue level established in EU (150 μg/kg, approx. 0.26 μM)."

Round 2

Reviewer 1 Report

The authors responded well to the comments and addressed all the issues raised by the reviewers. I suggest the acceptance of this manuscript. 

Reviewer 4 Report

The authors addressed almost all the comments and suggestions.